# Therapeutic itineraries of children after snakebites in the Brazilian Amazon: A thematic drawing-and-story study

Joseir Saturnino Cristino[1,2], Altair Seabra Farias[1,2], Erica Silva Carvalho[1,2], Jacqueline Sachett[1,2], Fan Hui Wen[3], Felipe Murta[1,2], Vinícius Azevedo Machado[1,2], Wuelton Monteiro [1,2,4]*

1 School of Health Sciences, Universidade do Estado do Amazonas, Manaus, Amazonas, Brazil,
2 Department of Teaching and Research, Fundação de Medicina Tropical Dr. Heitor Vieira Dourado, Manaus, Amazonas, Brazil, 3 Bioindustrial Center, Instituto Butantan, São Paulo, Brazil, 4 Duke Global Health Institute, Duke University, Durham, North Carolina, United States of America

* wueltonmm@gmail.com

## Abstract

### Background

Snakebite envenomations (SBEs) impose a significant burden on children living in the Brazilian Amazon. In this region, children are at a higher risk of long-term disabilities and death. Therapeutic itineraries refer to the paths individuals take to seek and manage their health, encompassing both formal and informal healthcare systems. Even with an increasing interest in involving children in qualitative research in health sciences, researchers generally neglect children as subjects capable of reporting on their health status. The aim of this study was to describe the healthcare itineraries of children presenting at a tertiary hospital in Manaus, Brazilian Amazon, for medical assistance after snakebites.

### Methods

A thematic drawing-and-story study was performed to explore the healthcare itinerary of children aged 4–12 years who were admitted with a diagnosis of SBE in a tertiary hospital in Manaus, Brazilian Amazon, from July 2022 to March 2024. Data was analyzed by deductive content analysis. Data collection involved drawing and storytelling based on the snakebite experience of the participant from the moment of the bite to hospital. Sample size was defined by saturation.

### Results

Thirteen (65%) boys and seven (35%) girls, with an average age of 8.7 years, were recruited. Most of them were accompanied to the hospital by their mothers (65%). Time to medical care ranged from 1 to 84 hours. Data analysis highlighted five key

**Data availability statement:** All relevant data are in the manuscript and its supporting information files.

**Funding:** This work was supported by the Conselho Nacional de Desenvolvimento Científico e Tecnológico (CNPq productivity scholarships to JS and WM), the Fundação de Amparo à Pesquisa do Estado do Amazonas (Fapeam 005/2022 - POSGRAD 2022/2023 to JSC; Fapeam 023/2022 – Iniciativa Amazônia +10 and Fapeam 017/2023 – PRONEM to WM). The funders had no role in study design, data collection and analysis, decision to publish, or preparation of the manuscript.

**Competing interests:** The authors have declared that no competing interests exist.

themes: 1) Identification and understanding of SBEs in the process of initial parental care; 2) Children's understanding of the SBE and their journey to care; 3) Children's experiences with SBEs and their exposure to them in the environment; (4) Use of therapeutic practices during the children's journey to care; and 5) Contingencies in the healthcare itinerary of the children. The initial response to SBE in children is marked by challenges in communication between them and adults, delaying proper care. The unexpected event is a traumatic experience for children, with intense pain and reactions such as fear. Fragmented itineraries significantly increase the time needed to access antivenom. In some cases, children try to take care of themselves, but parental care is still predominant.

## Conclusion

The experiences of a snakebite in children reinforce the need for public policies, such as specific educational interventions, aimed at promoting early recognition of signs, validating children's voices, and discouraging harmful practices. Strengthening culturally sensitive and child-focused strategies is important for public health, as it enables the transformation of long, fragmented and improvised therapeutic itineraries into more timely, safe, and effective care pathways for pediatric snakebite victims in the Amazon.

### Author summary

Snakebite envenomation represents a significant public health challenge in the Brazilian Amazon, particularly among children. Due to their smaller body size and limited access to emergency care in remote areas, children are at higher risk of complications and long-term disabilities. Despite the high burden of snakebites in children, little is known about how children experience and respond to a snakebite emergency. This study used a playful method involving drawing and storytelling to allow 20 hospitalized children to describe their journey from the moment of the bite to receiving medical care. Their stories revealed confusion, fear, pain and long delays in getting help. Many children were alone or without close adult supervision, and most relied on home remedies before reaching a hospital. Long travel distances and fragmented transport further delayed treatment. The study highlights that children's voices are often unheard in research and emergency care planning. By listening directly to them, we can better understand their needs and improve strategies for faster, safer and more empathetic care in future snakebite emergencies.

## Introduction

Snakebite envenomations (SBEs) represent a relevant public health problem in tropical and subtropical areas, especially in rural and difficult-to-access regions, such as

the Brazilian Amazon [1]. Global estimates indicate that approximately 5.4 million people are bitten by venomous snakes each year, resulting in approximately 2.7 million SBEs, between 81,000 and 138,000 deaths and 400,000 cases of permanent disability, including amputation, contractures, psychological disability, restricted mobility and extensive scarring, particularly in low- and middle-income countries [2].

In Brazil, approximately 30,000 SBEs are reported per year, approximately 15% of them in children [3]. Regional disparities in SBE incidence and rates of poor outcomes exist in the country, with its Amazon region being disproportionately affected [4]. In addition, among SBE cases, a higher proportion of children is reported in the Amazon [5]. Regardless of the noteworthy burden, the number of intensive care units and pediatricians is proportionately lower in this region [3]. Children tend to have higher SBE severity and greater complications due to their small body mass and same venom volume inoculated in comparison to adults [6]. Living in rural areas, antivenom underdosage and time to care >3 h are risk factors for SBE severity in children in Brazil [3]. In this context of poor access to medical care, children may evolve to complications such as secondary bacterial infection, extensive necrosis and compartment syndrome, resulting in long-term disabilities [5]. The loss of autonomy at such an early stage of life due to an SBE-related disability may deprive children of sensory and social experiences and of learning their future roles in the community [5].

The concept of a *therapeutic itinerary* refers to the set of processes through which individuals and their families organize decisions and strategies to respond to an illness, including successive or overlapping events across popular, folk, and professional health subsystems. Rather than a linear or predetermined sequence, it is an articulated chain shaped by social, cultural, emotional, and structural factors [7]. In Brazil, this approach has been applied to understand health-seeking behaviors in vulnerable contexts [7,8]. Applying this framework to SBEs is crucial to capture not only structural barriers but also the sociocultural dynamics and decision-making processes shaping children's experiences in the Amazon.

The three health subsystems of the therapeutic itinerary can be identified in adult patients with SBE in the Brazilian Amazon: i) The popular subsystem comprises self-care practices, with an underlying influence of the family and community on the subject's decision, which includes homemade remedies, blessings and prayers, and self-medication with industrialized drugs [9,10]; ii) The folk subsystem, prominent for Amerindian populations, is based on indigenous medicine and rituals led by shamans and other healers [11]; and iii) The professional component of the system is usually located in hospitals of the urban areas, where antivenoms and ancillary treatment are available [12,13]. Access to antivenom treatment requires considerable effort on the part of SBE patients, and it is marked by resistance to seeking medical assistance, great fragmentation of the itineraries with several changes of expensive means of transport, and low acceptability of the healthcare provided in some hospitals [14].

### Children's agency and healthcare itineraries

An interest in children and childhood has not always been present throughout the history of humanity. According to the French historian Philippe Ariès, children only became an object of concern for families, as well as being valued, after the advent of industrial society in the 17th century [15]. According to this author, until the Renaissance, children were seen as miniature adults, and that is why they worked in the same places and wore the same clothes as them. The child differed from the man only in size and strength, while other characteristics remained the same, including the ability to learn and execute everyday tasks. The end of the 17th century was considered a milestone in the evolution of feelings in relation to childhood, a time when they really began to talk about children's fragility and their peculiarities and worry about their moral formation and education.

In the early 20th century, scientists familiar with the evolutionary theory of Darwin began elaborating descriptions of human psychological development, including Freud's theory of psychosexual development, Vygotsky's sociocultural theory of child development, Erikson's psychosocial theory, and Piaget's and Wallon's psychogenetic theories [16]. Innovative psychological and epistemological contributions were brought about by these theories. From Piaget's studies, for example, it is concluded that children are not simply less developed versions of adults, but they think and perceive the world in

a qualitatively different way; children are not simple passive receivers of information, but are in constant interaction with the environment, actively building their understanding of the world; and that the most effective way to understand children's reasoning is to approach problems and issues from their point of view [17].

In the field of anthropology, the most famous studies that have children as their focus are those carried out in 1920s and 30s by North American anthropologists linked to the School of Culture and Personality, especially those of Margaret Mead. These anthropologists were concerned with understanding what it means to be a child or adolescent in other sociocultural realities, often taking North American society at the time as a counterpoint [18,19]. These studies are marked by the division between adulthood and childhood and refer to an idea of development of the mature personality, in which the adult is the ultimate end of the developmental process. A second moment in this area is given by the British Structural-Functionalist School, in which children are mere receptacles of functional roles that they play throughout the socialization process, in which their actions and symbolic representations are given by the system itself [20].

From the 1960s onwards, when culture began to be understood as a symbolic system, the idea that children gradually incorporate culture during their maturation was revisited. According to Brazilian anthropologist Clarice Cohn [21], the difference between children and adults is no longer quantitative, but qualitative – '*children do not know less things, they know different things*'. Child anthropology is no longer confused with analyses of cognitive development; on the contrary, it dialogues with them. Children are not only produced by cultures, but producers of culture, actively engaging with adults and other children in the constitution of social relationships. This paradigm shift has relevant methodological implications, since children are now seen as independent subjects in their decisions and narratives and are active informants in research [22,23].

With an increasing interest in involving children in qualitative research [24,25], a recent scoping review proposed the use of the term *healthcare itinerary* for children, expanding the original concept of *therapeutic itinerary*. By adopting this terminology, the concept is no longer restricted to illness, but encompasses the entire process of living, preventing illness, becoming ill, seeking care, and recovering within the different health subsystems [8]. Children's healthcare itineraries are generally described based on information provided by their caregivers [8]. Most of these publications describe therapeutic and rehabilitative itineraries, with the use of both professional and popular healthcare sector. Studies of emergency itineraries involving children are scarce in the literature, mostly focusing on delays in prehospital aid and specialized care [26–29].

The aim of this study was to describe the healthcare itineraries of children presenting at a tertiary hospital in Manaus, Brazilian Amazon, for medical assistance after SBEs, using a participatory approach from children's drawings and narratives.

## Methods

### Ethics statement

The data collection for this study was carried out after approval by the Human Research Ethics Committee of the Fundação de Medicina Tropical Doutor Heitor Vieira Dourado (FMT-HVD) (approval number (#5.286.460/2022). All participants' legal guardians signed a consent form. After a detailed explanation of the project, literate children signed an assent form to participate in the study. Non-literate children provided verbal assent. All participants who were approached agreed to take part in the study.

### Research team and reflexivity

In-depth interviews and the drawing method were conducted by a MSc-level male researcher with extensive experience in qualitative research and care of SBE patients (J.S.C.), with a male PhD-level qualitative researcher who has a background in Pedagogy and experience in educating children, acting as an observer (V.A.M.). The interviews and drawing method guides were developed by J.S.C., V.A.M., J. S., F.M. and W.M. Data analysis was conducted by J.S.C., V.A.M.,

A.S.F., F.H.W. and W.M. The study team had no prior relationship with the participants. All the team members have previously carried out qualitative research with SBE patients and health professionals in the Brazilian Amazon. The funders of the study had no role in the study design, data collection, data analysis, data interpretation or writing of the manuscript.

The research team acknowledged the potential power dynamics of interviewing children in a hospital setting and sought to minimize them by adopting a child-centered approach, using drawings as a participatory tool, avoiding white coats, employing clear and informal language, and ensuring a calm and private environment. Parents were first approached separately to reduce undue influence. Cultural conceptions of SBEs in the Amazon were respected, with attention to local beliefs and practices. This reflexive stance aimed to reduce potential biases and preserve the authenticity of the children's voices.

### Participants

This is a qualitative study carried out at the FMT-HVD, a tertiary hospital for SBE treatment located in Manaus, Western Brazilian Amazon. A consecutive sample of 20 snakebite patients aged 4–12 years were enrolled from July 2022 to March 2024. We used the method of theoretical saturation, defined as the point at which no new elements relevant were identified [30].

This age group includes the late preoperational (4–7 years, when children begin to think symbolically and learn to use words and pictures to represent objects and are getting better with language and thinking) and the concrete operational stages (7–12 years, when children begin to think logically about concrete events, thinking becomes more logical and organized, and they begin using inductive logic, or reasoning from specific information to a general principle) of cognitive development [17]. In the late preoperational stage (schematic drawing stage), children will have already developed a simple system for communicating their ideas by drawing and individual shapes are consistently used to represent objects. In the concrete operational stage (realistic drawing stage), motor skills and spatial awareness will have improved, with the schema from the earlier stages becoming more complex. Children will discuss their art more freely and can even be self-critical as they strive for greater realism [31].

### Study design

All the children included in the study were admitted at FMT-HVD for SBE treatment during the study period. Recruitment occurred when a member of the hospital's staff notified the research team upon the arrival of a child with SBE at the emergency department. The research team first approached the parents and, after obtaining their approval, explained the study to the child and asked whether they were willing to participate.

On admission, SBEs were diagnosed based on their clinical and epidemiological characteristics and, when the patient brought the snake that caused the envenomation, by the identification of the perpetrating snake, which was done by a trained biologist using a validated taxonomic key for the study region [32]. Upon consent by the parents and assent by the children, the study team collected the clinical and epidemiological data of the case and performed the qualitative investigation of the healthcare itinerary.

In this study, a health itinerary was defined as the set of processes by which children and their caregivers choose, evaluate, and adhere to different forms of treatment, including sociocultural practices for coping with illness [7]. For the researcher, it represents the patient's individual interpretation of the illness experience, a conscious attempt to restore health and bring coherence to the diverse and dispersed actions undertaken along the path. We assume that the children's health itineraries in this study may involve three subsystems: the popular, the folkloric, and the professional [33].

Patients were treated according to the Brazilian Ministry of Health guidelines [34]. All the interventions were provided at no cost to the patients.

Study flowchart and data collection procedures are presented in Fig 1.

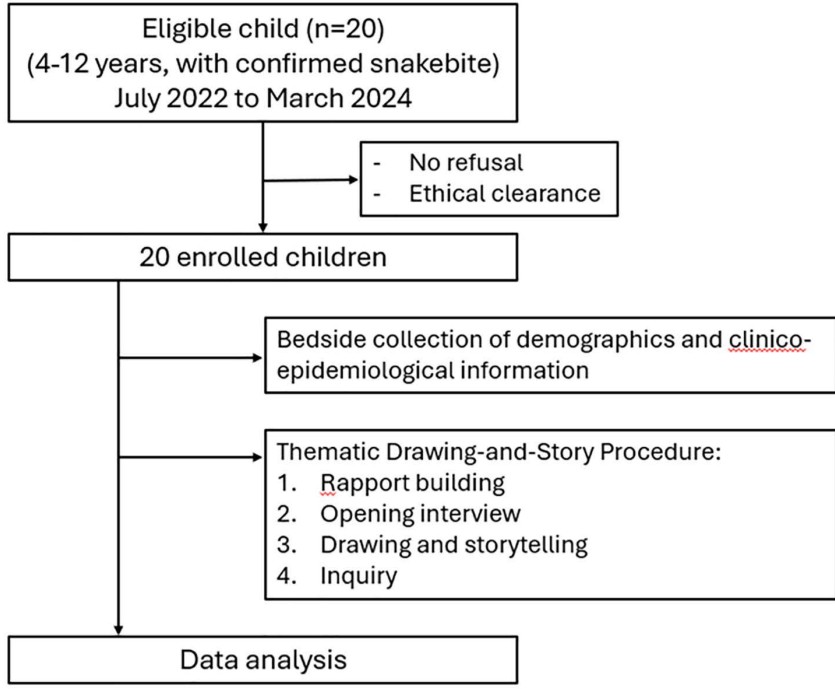

**Fig 1. Study flowchart and data collection procedures.**

## Data collection

*Epidemiological and clinical data:* Epidemiological and clinical data were collected at the bedside during the hospital stay using a standardized form. Baseline data included information on gender, age (in years), area of occurrence (urban or rural), ethnic background, years of schooling, perpetrating snake species, anatomical site of the bite, time elapsed from bite to medical assistance (in hours), and pre-admission treatments (use of topical or oral medicines, use of tourniquet and other procedures). Information on clinical manifestations was also collected. All demographic, clinical and laboratory information was collected via a standardized clinical registration form (REDCap, Vanderbilt University).

*Thematic drawing-and-story procedure (TD-SP)*: TD-SP is a tool used in qualitative research that involves drawing and storytelling based on a theme [35]. The method can be used with people of any age, group or socio-economic and cultural level, if they can draw and verbalize. In this study, children were approached for TD-SP 12–24 hours after hospital admission, after antivenom treatment (if necessary) and pain control. First, the researchers introduced the project to the children and, after rapport building, the opening interview with the following five questions was performed: *1) What happened to you? 2) What did you feel at the time of the snakebite? 3) Where did the bite happen? 4) What were you doing at that time? 5) How did you get here?*

In the phase of drawing and storytelling, the researchers asked the participants to draw "*the path you took from the time the snake bit you until you arrived at the hospital*" on an A4 (21 × 29.7 cm) sheet of white drawing paper using a set of 12 crayons and a pencil. There was no time limit. Erasers or text correctors were not provided. In the inquiry phase, they were then asked to provide a verbal narrative recounting story behind the drawings. The children were not asked to paint or describe a specific experience that had happened to them. However, in case of discomfort, children were assured of getting preliminary assistance from the research team by referring them to the hospital psychologists. Participants were assured that their identities would remain confidential, and their drawings and narratives would be kept anonymous. Parents were assured that the analysis of the drawings and the narratives would be conducted as a group and that

the drawings would be used for research purposes. The TD-SP lasted approximately 1 hour and took place in a quiet, comfortable room and was recorded for later transcription. The interviewer and the observer took field notes. Eighteen participants spoke Portuguese fluently, and one was fluent in Apurinã and another in Waimiri, both Amerindian languages of the Brazilian Amazon. For these participants, the mothers served as the interpreter, since they are a bilingual Portuguese-Apurinã and Portuguese-Waimiri speakers. The interviews were transcribed and deidentified. Potential bias was minimized considering the reflexivity process above described. All the interviews were recorded using a digital voice recorder (Sony ICD PX240 4GB). The interviews were transcribed to a file in Microsoft Word. The results and methods report follows the Consolidated Criteria for Reporting Qualitative Research (S1 File). The characteristics of the children enrolled in the study are presented in S2 File.

### Data analysis

Epidemiological and clinical data were presented as descriptive statistics. For the descriptive analysis of the itineraries, we adopted the classification proposed by Cristino et al. [8] according to decision-making agent, presence of a companion during the itinerary, use of the healthcare subsystems (folk, professional or popular), continuity of the physical itinerary, perception of the effectiveness of the care, severity of the health problem, means of transport used, planning and completeness of the itinerary.

The style and content of the drawings were detailed descriptively based on Goldner et al. [36]:

*Drawing style indicators:* (1) drawing type (scribbling, pre-schematic, schematic, realistic, no drawing), (2) whether the participant is depicted in the drawing (yes/no), (3) whether the perpetrating snake is depicted in the drawing (yes/no), (4) physical contact or short distance between the victim and the snake, (5) the presence of the companion on the itinerary (yes, no), (6) the means of transport used (yes, no), (7) the route taken to the hospital (yes, no), (8) the hospital (yes, no), (9) number of scenes represented in the drawing (one, two or more), (10) the drawing conveys the idea of displacement or dynamism of the elements, and (11) whether the drawing included words (yes/no).

*Content indicators:* (1) depiction of a painful scene in the drawing (yes/no), (2) depiction of the bite injury in the drawing, such as fang marks, blood, swelling, dressing on the limb (yes/no), (3) depictions of systemic injury manifested in bleeding, headache, fever or others (yes/no), (4) depictions of receiving care, such as medicines or dressings, during the itinerary (yes/no).

Narratives were transcribed into a Word file. A pre-analysis phase was conducted, which involved reading of the interviews and condensing the textual data into a summarized format. Field notes and raw data from the transcribed narratives were imported into the Atlas.ti software (version 7.3.1). The second phase was represented by the method triangulation [37], generating a combined data set with interviews, drawings and clinic-epidemiological characteristics, to establish clear links between the research objectives and the summarized findings derived from the raw data [38]. The categories emerged through inductive content analysis and were discussed among the researchers for consensus. Drawings and audio recordings of the narratives were separated according to each interviewee to represent the emerging themes. Finally, the findings were discussed based on relevant existing literature on therapeutic itineraries and related subjects. The analysis was concluded in December 2024.

## Results

### Characteristics of the participants

Table 1 shows the demographic and clinical characteristics of the study participants. Of the 20 participants, 13 were male (65%). The most frequent age groups were 11–12 years old (35%), 4–6 years old (25%) and 9–10 years old (25%). Regarding ethnicity, 16 were *pardo* (80%) and four were indigenous (20%). The majority live in rural areas (75%). Bites occurred on the lower limbs, with a predominance of the feet (50%). All the SBEs were diagnosed as being caused by *Bothrops atrox*, the Amazonian lancehead (popularly named *jararaca* or *surucucu* in Portuguese).

**Table 1. Characteristics of the 20 study participants.**

| Variable | n (%) |
|---|---|
| **Gender** | |
| Male | 13 (65%) |
| Female | 7 (35%) |
| **Age groups (years)** | |
| 4-6 | 5 (25%) |
| 7-8 | 3 (15%) |
| 9-10 | 5 (25%) |
| 11-12 | 7 (35%) |
| **Ethnicity** | |
| *Pardo*[1] | 16 (80%) |
| Indigenous[2] | 4 (20%) |
| **Area of occurrence** | |
| Rural | 15 (75%) |
| Urban | 2 (10%) |
| Peri-urban | 3 (15%) |
| **Affected region** | |
| Foot[3] | 10 (50%) |
| Ankle[3] | 5 (25%) |
| Toe | 4 (20%) |
| Knee | 1 (5%) |
| Leg | 1 (5%) |
| **Perpetrating snake** | |
| *Bothrops atrox* (Amazonian lancehead) | 20 (100%) |
| **Formal education**[4] | |
| No | 1 (5%) |
| Preschool | 4 (20%) |
| Elementary school (years) | |
| 1-2 | 1 (5%) |
| 3-4 | 8 (40%) |
| 5-6 | 2 (10%) |
| 7-8 | 4 (20%) |
| **Time to medical care (hours)** | |
| <2 | 6 (30%) |
| 2-6 | 7 (35%) |
| 7-12 | 3 (15%) |
| >20 | 4 (20%) |
| **Case severity on admission**[5] | |
| Dry bite[6] | 1 (5%) |
| Mild | 16 (80%) |
| Moderate | 2 (10%) |
| Severe | 1 (5%) |
| **Antivenom treatment** | |
| Yes[7] | 19 (95%) |
| **Clinical manifestations in the bitten limb** | |
| Pain | 20 (100%) |
| Edema | 19 (95%) |

*(Continued)*

**Table 1.** (Continued)

| Variable | n (%) |
|---|---|
| Secondary bacterial infection | 12 (60%) |
| Compartment syndrome | 1 (5%) |
| Necrosis | 1 (5%) |
| **Systemic manifestations** | |
| Ecchymosis | 1 (5%) |
| Epistaxis | 1 (5%) |
| Hematuria | 1 (5%) |
| **Hospitalization (days)** | |
| 1-3 | 7 (35%) |
| 4-8 | 8 (40%) |
| 9-15 | 4 (20%) |
| 40 | 1 (5%) |

[1]In Brazil, *pardo* is an ethno-racial and skin color category used by the Brazilian Institute of Geography and Statistics in the Brazilian censuses. The term used to refer to Brazilians of mixed ethnic ancestries, with a diverse range of skin colors and ethnic backgrounds.

[2]Two of the Waimiri-Atroari, one of the Kambeba, and one from the Apurinã indigenous groups.

[3]One patient had multiple bites on his feet and ankles.

[4]In Brazil, preschool is a stage of basic education that is given to children aged 4–5 years old. Elementary education is given to children aged 6–14 and consists of nine years of schooling. The Brazilian Constitution dictates that both are mandatory and free of charge.

[5]As defined by the official guideline of the Brazilian Ministry of Health.

[6]Dry bites are characterized by the absence of venom being injected into the victim during a bite by a venomous snake. Fang marks were observed, but without clinical signs of envenomation.

[7]The patient diagnosed with a dry bite was kept under observation in the hospital for 24 hours and did not receive antivenom, as he did not show clinical signs of envenomation during this time.

Fifteen participants were in elementary school (75%) and four in preschool (20%). Thirteen participants received medical care within six hours of the bite (65%), and four received medical care more than 20 hours after the bite (20%). On admission, 80% of cases had a mild degree of severity. Only one participant did not receive antivenom, as it was diagnosed as being a dry bite. All the patients presented pain, and 19 (95%) presented edema in the affected limb. Twelve patients developed secondary bacterial infection (60%). One severely ill patient, who had multiple fang marks and who received medical care 84 hours after the bite, developed necrosis and compartment syndrome. Minor systemic bleeding was observed in three participants. Hospitalization time varied from one to forty days.

The most frequent environment in which the snakebite occurred was the peridomestic area (40%), followed by work areas (25%) and narrow trails used to access other residences or small shops in the rural areas (20%). Two children were bitten inside their houses (10%).

At the time of the snakebite, the children were helping their parents with external professional activities or household chores, playing indoors or in the peridomestic area, collecting fruits in the backyard, or walking along trails routinely used in the communities (S3 File).

### Classification of therapeutic itineraries

Table 2 presents the classification of the therapeutic itineraries of the study participants. The itineraries were predominantly passive (90%), except for P1 and P6, classified as mixed itineraries. P1, after the snakebite, applied a homemade remedy prepared by his mother (alcohol with herbs), and P6 performed a tourniquet, both without adult guidance. These

**Table 2. Classification of the therapeutic itineraries of the 20 study participants.**

| Base for classification | Type of itinerary[1] | n (%) |
|---|---|---|
| Decision maker | Passive | 18 (90%) |
| | Mixed | 2 (10%) |
| Presence of a companion during the itinerary | Accompanied[2] | 19 (95%) |
| | Mixed | 1 (5%) |
| Use of healthcare sectors | Multidimensional[3] | 18 (90%) |
| | Unidimensional | 2 (10%) |
| Continuity of the itinerary | Fragmented | 18 (90%) |
| | Continuous | 2 (10%) |
| Child's perception of the effectiveness of care | Effective | 1 (5%) |
| | Non reported | 19 (95%) |
| Severity of the health problem | Emergency | 20 (100%) |
| Administration of the healthcare provider | Public | 20 (100%) |
| Means of transport used | Land | 12 (60%) |
| | Mixed[4] | 8 (40%) |
| Itinerary's reporter | Self-reported | 20 (100%) |
| Planning of the itinerary | Planned | 2 (10%) |
| | Improvised | 18 (90%) |
| Itinerary's completeness | Complete | 20 (100%) |

[1]For details on the typology of health itineraries see Cristino et al. [8].

[2]Two participants (P5 and P6) performed part of the itinerary in the company of another minor. P19 performed part of the itinerary alone.

[3]Combination of the popular and professional sectors of healthcare.

[4]Many combinations of land, river and air transport means.

actions represented isolated active decisions by the children; however, in all subsequent steps of their therapeutic itineraries, decision-making was passive and carried out by their parents or other companions.

Nineteen children were accompanied by their caregivers throughout the entire itinerary. Among them, two also received assistance from other minors at the beginning of the journey. Only one child had the itinerary classified as mixed, as part of the trajectory was undertaken alone before receiving help from another person. The two children who received support from other minors were: P5, who was initially with a 7-year-old cousin, being carried by him to a neighbor's house to get help, and P6, who at the beginning, was with a 14-year-old teenage brother, who contacted his mother by phone. The only mixed itinerary was P19, who suffered the bite while she was alone at home, cleaning the house and yard, and taking care of two brothers (one three years old and one eight months old), as requested by her mother. The participant hid the fact for hours, and only mentioned the problem to her mother hours later, when she was already experiencing intense pain and hematuria.

Eighteen participants performed multidimensional itineraries, which included popular treatments and professional care. In most cases, the route was fragmented (90%), and included land, air and river transport means. P4 and P8 followed a continuous and direct route to the hospital. These two were also the only ones to have a planned itinerary, knowing in advance the destination for receiving hospital care. In his narrative, only P19 reported a perception of improvement after treatment with antivenom. All routes were considered as emergencies and the participants were treated exclusively by the public health system.

The average time from snakebite to hospital admission was 11.2 hours, with a duration ranging from 1 to 84 hours. The route is usually made up of several fragments, represented using several means of transport and waiting times that can reach several hours to find another means of transport to continue the journey to the hospital (Fig 2). Transportation was

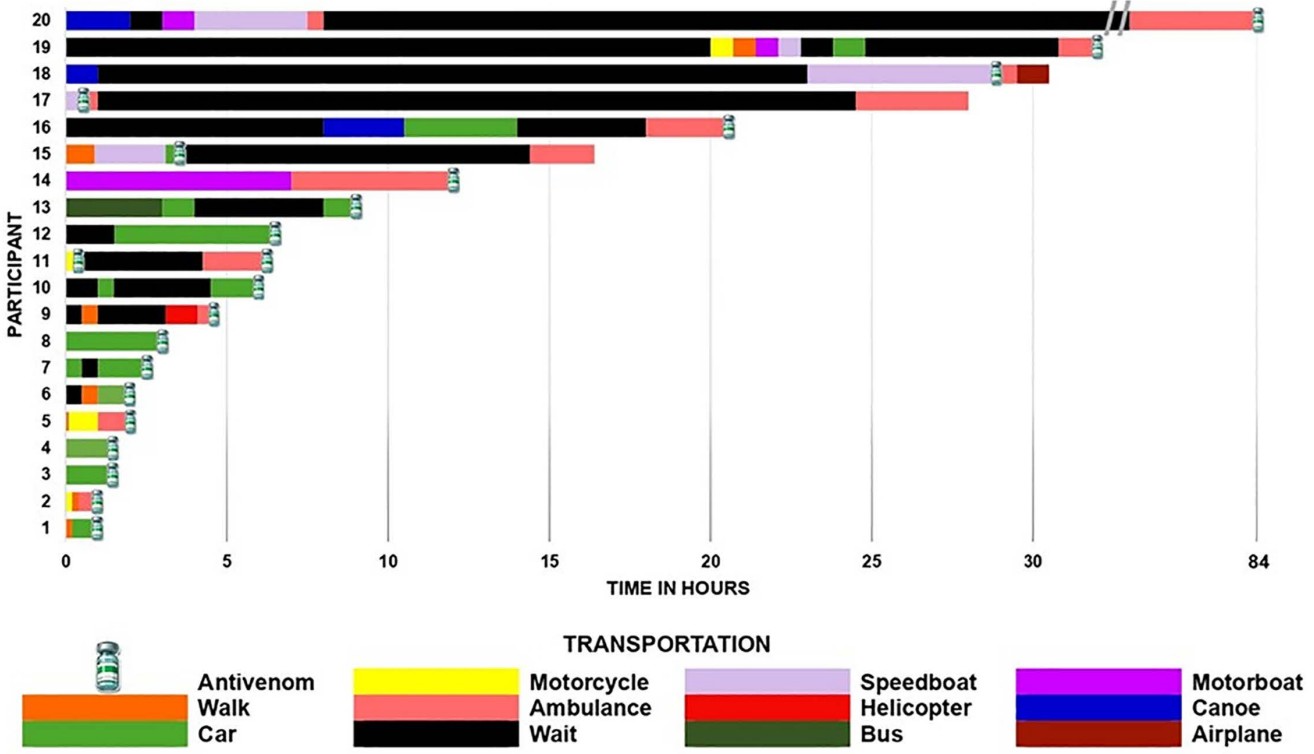

**Fig 2. Itineraries of the study participants expressed graphically for the time from the snakebite to hospital admission.** The different colors of the fragments of each itinerary represent the means of transport used along the route.

predominantly by land (60%), but eight children used multiple means of transportation, including speedboats, canoes, motorboats and helicopters. Four (20%) of the twenty children (P11, P15, P17 and P18) received antivenom in countryside municipalities before admission. However, due to limited support and specialized care, they were transferred to Manaus. P11 received an additional dose of antivenom at the referral hospital, since the signs and symptoms had not yet improved.

The itineraries were entirely self-reported by the children. All were completed without interruptions on the way to hospital care.

## Drawing analysis

The analysis of the drawings showed variation in type, style, and content. Schematic and realistic drawings contained a greater number and diversity of elements, while simpler drawings presented fewer details. A detailed breakdown of the indicators is presented in Table 3.

## Emerging themes

Five key themes were derived from the content analysis of participants' interviews and drawings: 1) Identification and understanding of the snakebite in the initial parental care; 2) Children's understanding of the snakebite and their healthcare itinerary; 3) Children's experience with the snakebite and environmental exposure; 4) Use of popular therapeutic practices in the children's healthcare itinerary; 5) Contingencies during the children's healthcare itinerary.

**Table 3. Results of the style and content indicators from the drawing analysis.**

| Drawing indicators | Response | Drawing types | | | | |
|---|---|---|---|---|---|---|
| **Style indicators** | | **Scribbling (n=2)** | **Pre-schematic (n=2)** | **Schematic (n=10)** | **Realistic (n=6)** | **Total (n=20)** |
| Depiction of the participant in the drawing | Yes | 0 | 2 | 8 | 6 | 16 (80%) |
| | No | 2 | 0 | 2 | 0 | 4 (20%) |
| Depiction of the perpetrating snake in the drawing | Yes | 0 | 1 | 7 | 4 | 12 (60%) |
| | No | 2 | 1 | 3 | 2 | 8 (40%) |
| Physical contact or short distance between the victim and the snake | Yes | 0 | 1 | 5 | 3 | 9 (45%) |
| | No | 0 | 0 | 1 | 1 | 2 (10%) |
| | Absent element[1] | 2 | 1 | 4 | 2 | 9 (45%) |
| The presence of a companion on the itinerary | Yes | 0 | 2 | 5 | 5 | 12 (60%) |
| | No | 2 | 0 | 5 | 1 | 8 (40%) |
| Depiction of the means of transport used | Yes | 0 | 0 | 7 | 4 | 11 (55%) |
| | No | 2 | 2 | 3 | 2 | 9 (45%) |
| Depiction of the route taken to the hospital | Yes | 0 | 0 | 4 | 3 | 7 (35%) |
| | No | 2 | 2 | 6 | 3 | 13 (65%) |
| Depiction of the hospital | Yes | 0 | 0 | 3 | 3 | 6 (30%) |
| | No | 2 | 2 | 7 | 3 | 14 (70%) |
| Number of scenes represented in the drawing | Two or more | 0 | 0 | 4 | 3 | 7 (35%) |
| | One | 2 | 2 | 6 | 3 | 13 (65%) |
| The drawing conveys the idea of displacement or dynamism of the elements | Yes | 0 | 0 | 4 | 3 | 7 (35%) |
| | No | 2 | 2 | 6 | 3 | 13 (65%) |
| The drawing included words | Yes | 0 | 2[2] | 1[2] | 1[3] | 4 (20%) |
| | No | 2 | 0 | 9 | 5 | 16 (80%) |
| **Content indicators** | | | | | | |
| Depiction of a painful scene in the drawing | Yes | 0 | 0 | 1 | 1 | 2 (10%) |
| | No | 2 | 2 | 9 | 5 | 18 (90%) |
| Depiction of the bite injury in the drawing, such as fang marks, blood, swelling, dressing on the limb | Yes | 0 | 0 | 1 | 0 | 1 (5%) |
| | No | 2 | 2 | 9 | 6 | 19 (95%) |
| Depictions of systemic injury manifested in the form of bleeding, headache, fever or others | Yes | 0 | 0 | 0 | 0 | 0 (0%) |
| | No | 2 | 2 | 10 | 6 | 20 (100%) |
| Depictions of receiving care, such as medicines or dressings, during the itinerary | Yes | 0 | 0 | 1 | 1 | 2 (10%) |
| | No | 2 | 2 | 9 | 4 | 17 (85%) |

[1]Participant and/or snake not depicted in the drawing.

[2]Three participants signed their name on the drawing.

[3]One participant noted the time of the snakebite on the drawing.

## 1) Identification and understanding of the snakebite in the initial parental care

After the snakebite, when asking for help, many children received immediate attention from parents, siblings, cousins, neighbors, and even strangers who happened to be nearby. After the traumatic incident, some children managed to run away from the snake, while others received help to kill it. However, not all assistance occurred immediately. Participant 5 was initially helped not by an adult, but by his cousin, a child only 10 years old, who carried him to a neighbor's house in search of help. Participant 6 was home alone with his 14-year-old brother at the time of the incident, the brother called their mother to assist them. Participant 9 had to wait until his father arrived a few minutes later and, upon seeing him crying, provided the first care. Participant 16 received immediate attention from his father and uncle, who were with him at the time, but since neither of them had seen the snake, they decided to wait. It was only the following day, after the symptoms worsened, that they took him for medical attention. Even more critically, participant 19 only informed her mother about the incident after realizing she was urinating blood.

> *"I quickly pulled my foot away and ran. Then, I stayed there crying, and my dad called a man who came to get me on a motorcycle" (P2, 11 years old, male)*

> *"My uncle found the snake and chopped it into four pieces" (P4, 8 years old, female)*

> *"I had no strength in my leg. I couldn't walk. My cousin had to carry me, so he carried me to my neighbor's house" (P5, 12 years old, female)*

> *"I called out to my older brother [another minor, 14 years old] […] He was on the phone inside the house" (P6, 7 years old, male)*

Initially, for the adults who did not witness the moment of the incident, it was challenging to understand the child's narrative and identify the signs of a snakebite, leading them to seek facts that would corroborate what the child was saying. This difficulty in understanding on the part of the adults may have occurred due to the common belief that children have a limited ability to judge what has happened to them, as observed in accounts where adults needed to visually confirm the presence of the snake to believe the child's report. Moreover, the natural difficulty children have in describing the event may have contributed to this disbelief regarding their cognitive competence and the truthfulness of their accounts.

> *"Mommy asked: "Sweetie, are you sure it was a snake?" I said: "Yes, mommy! I saw it, I saw the snake, I saw it" (P1, 9 years old, female)*

> *"Daddy got mad because I was crying... He didn't think it was a snake" (P9, 5 years old, male)*

> *"Dad, something bit me. Then, I started crying. My uncle turned and said: "That's definitely a "formiga catalão" bite ["formiga catalão" refers to a type of fire ant]" (P16, 10 years old, male)*

> *"My mom saw it, she has a picture of my pee that looks like coffee [...] then we went to sleep, and only the next day when we woke up, we went [referring to seeking medical help]" (P19, 9 years old, female)*

## 2) Children's understanding of the snakebite and their healthcare itinerary

The snakebite was perceived as unexpected by the children, who did not imagine that such an event could occur and were unable to determine where the snake had come from. In the drawings and narratives, it was observed that, at the moment of the accident, the children depicted themselves engaged in regular daily activities, such as playing or even accompanying their parents or guardians in work-related tasks like fishing, fruit gathering, and even house repairs, as in the case of participant 8. In these settings, the children did not notice the presence of the snake until they felt the pain of

the bite (Figs 3 and 4). Upon experiencing the snakebite, they expressed feelings of fright, despair and fear, often conveyed in their accounts through crying and a lack of facial expression in their drawings. During the event, they always called someone nearby for help.

The clinical signs reported by the participants were mainly related to the bite site, such as redness, bruising and swelling; however, there were cases with systemic complications, including one child who reported hematuria and another who required surgical procedures such as fasciotomy and removal of necrotic tissue. The symptoms mentioned primarily included throbbing pain and burning at the bite site, as well as muscle weakness in the lower limbs, drowsiness and temporary shortness of breath.

*"In the ambulance on the way here, I felt a bit short of breath, but when I got to the hospital it got better"* (P2, 11 years old, male)

*"I had no strength in my leg, I couldn't walk. My cousin had to carry me"* (P5, 12 years old, female)

*"I jogged with my foot lifted because I couldn't walk, it was hurting"* (P16, 10 years old, male)

*"I think it was around two in the morning when I woke up to pee. Then, I think my pee was very yellow, tea-colored, but very strong. Then, I peed again in the morning, when I woke up, and I was peeing blood"* (P19, 9 years old, female)

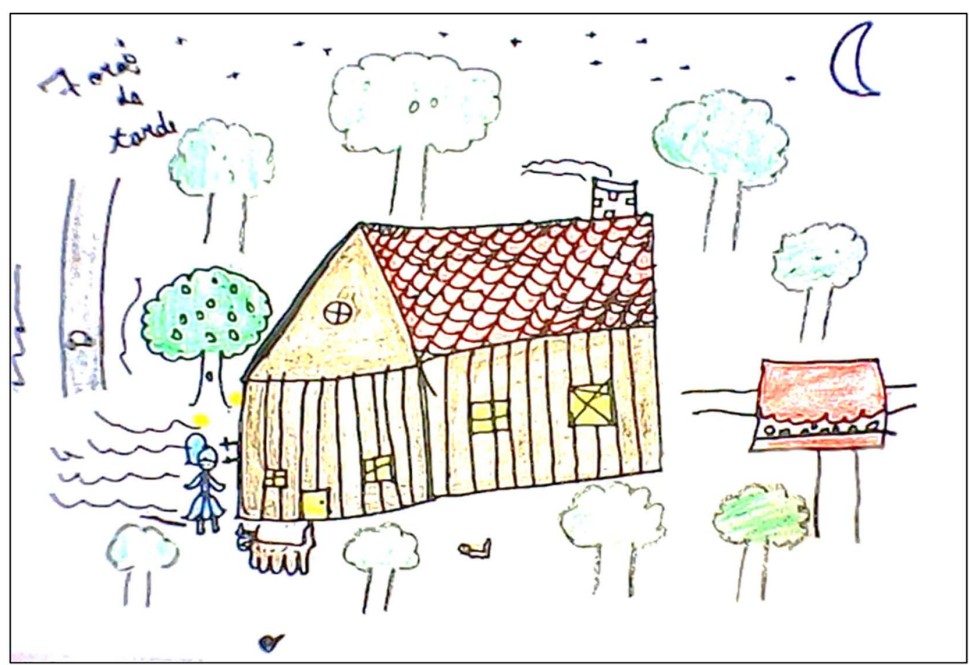

**Fig 3. Drawing made by participant 1 depicting the moment when the snakebite occurred.** The scene portrays the bite suffered by a 9-year-old female child, a 3ʳᵈ-grade elementary school student, while playing at night in the backyard of her home, accompanied by her two dogs, one larger and one smaller. The incident occurred in the peridomestic area, specifically in the backyard, as shown in the child's drawing and reported in the verbal account. While playing, the girl was bitten by a snake, which she herself illustrated in her visual report, placing the coiled animal at a certain distance, in the lower area of the drawing. Although the snake was not depicted in direct contact with the victim, its presence was clearly identified, even if represented in a small and peripheral way within the visual space. The drawing contains realistic features appropriate for the child's age, including details of color, texture and proportion that are close to real-life references, demonstrating good observational and representational ability. After the incident, the child was immediately carried into the house and later taken by her parents in their own car to a pediatric emergency service in the city of Manaus (Amazonas), and subsequently to the Dr. Heitor Vieira Dourado Tropical Medicine Foundation (FMT-HVD). Before leaving the house, the parents searched for the snake and prepared the car, while the child, on her own initiative, applied a homemade remedy to the bite. However, the journey to the hospital, the means of transportation used, and the medical care received were not represented in the drawing. The child included herself in the scene, placing her figure peripherally within the visual space. Although pain, local injury or systemic symptoms were not illustrated, the drawing includes key elements of the moment of the incident, such as the nighttime setting with the moon and stars in the sky. Words were also observed in the image, including the approximate time of the incident.

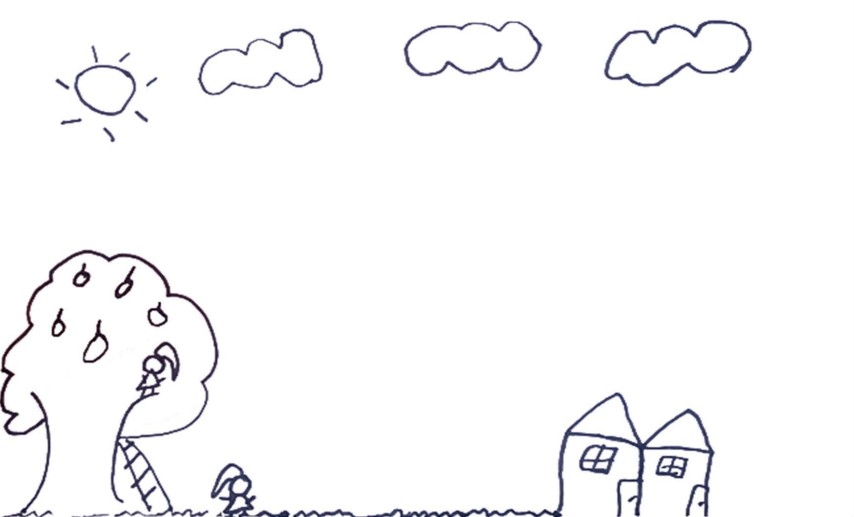

**Fig 4. Drawings made by participant 4 depicting the moment when the snakebite occurred.** The scene represented refers to the bite suffered by an 8-year-old female child, a 3rd-grade elementary school student. The incident occurred in the peridomestic area, i.e., near her grandmother's house, while she was collecting bacuri fruit in the backyard, accompanied by her sister. The bite occurred during the day, in a familiar and sunny environment. Although the child did not see the snake at the moment of the bite, only felt it, she was immediately taken by car to the Dr. Heitor Vieira Dourado Tropical Medicine Foundation (FMT-HVD), accompanied by her mother. In the drawing produced by the child, the episode is depicted in a simple manner, based on her memory and perception of the incident. The snake is not shown in the drawing, as the animal was not seen, but other important elements are included. On the left side of the drawing, there is a bacuri tree with hanging fruits and a ladder connecting the ground to the treetop, indicating the fruit-collecting context. The child's sister is shown at the top of the tree, while the child herself is represented at ground level, standing over low vegetation, where the bite occurred. The environment in the drawing includes a clear sky with a shining sun and scattered clouds, conveying a sense of calm. The grandmother's house is also depicted near the site of the incident, suggesting both proximity and a familiar setting. The drawing follows a schematic style appropriate for the child's age, with organized and recognizable elements such as houses, trees, clothing and figures. The child is positioned peripherally in the scene, and the drawing demonstrates good graphic representation skills for her age. The snake, hospital care, the route to the hospital, and the means of transportation used are not represented in the drawing. Nor are pain, wounds or symptoms associated with the bite depicted. The child focused on reconstructing the environment and actions at the moment of the incident, rather than its clinical or emotional outcomes.

The children mostly followed an improvised and fragmented path (90%). In addition to not knowing the first steps to take after a snakebite, such as washing the wound and immediately seeking professional care, their guardians, upon realizing what had happened, also did not know which healthcare facility they should go to. They headed toward any nearby health unit, also unaware of the appropriate care and destination in the case of a snakebite (Figs 5 and 6).

### 3) Children's experience with the snakebite and environmental exposure

Children who have had previous experience with snakebites, such as living in environments where snakes were commonly present, witnessing their parents kill snakes nearby, or hearing accounts from neighbors who had suffered such incidents, demonstrated a sense of confidence and safety in the environment they were in, as these were familiar places they visited daily. This led to a lack of preventive care, with no protective measures being taken and the possibility of such an incident being ignored.

*"Near my friend's house, they've already killed two snakes there. And inside another house, on the way to the bakery, they killed two as well... But I never imagined the snake would bite me" (P5, 12 years old, female)*

*"We were playing with the chicken and the dog, but the chicken ran away really fast after the snake bit me" (P12, 5 years old, female)*

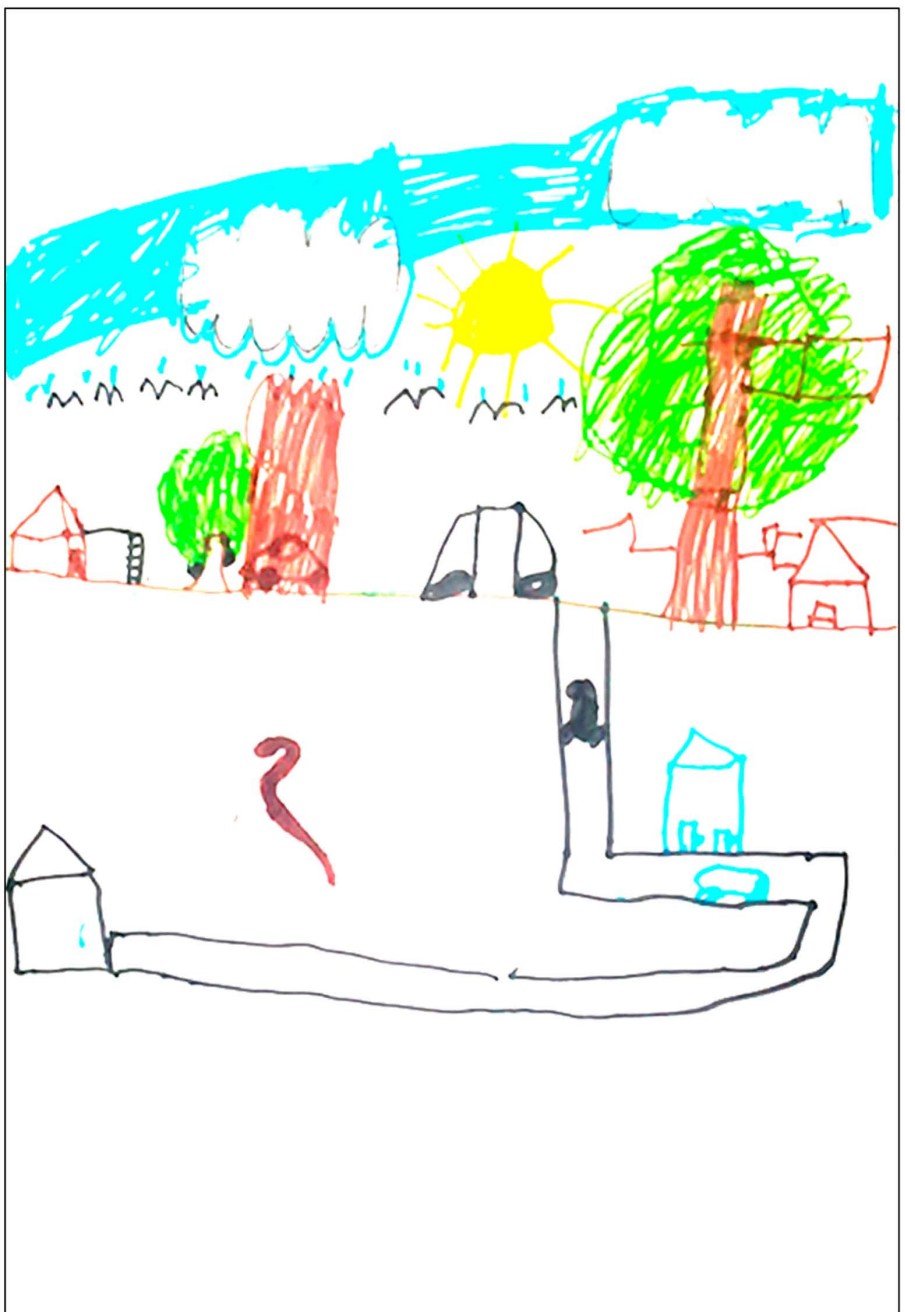

**Fig 5. Drawings made by participant 6 showing the journey undertaken.** The drawing portrays a snakebite experienced by a 7-year-old male child, a 3rd-grade elementary school student. The incident occurred in the peridomestic area, specifically in the backyard of his home, while the boy was on his way to the outdoor bathroom. After being bitten and waiting for his mother to return home, he was carried by her through the rain over a long distance along a dirt track to the nearest road, where they managed to get a ride from a driver who happened to be passing by at that moment. The boy was taken to an emergency care unit in the city of Manaus and later transported in another private vehicle to the Dr. Heitor Vieira Dourado Tropical Medicine Foundation (FMT-HVD). The drawing created by the child presents a well-organized visual narrative that represents the main events related to the incident. On the left side of the composition, one can observe the domestic setting, with large trees, the house, a blue sky with clouds, the sun, and even rain, portraying the environment and time of the incident. At the center of the image, the snake, drawn in red and exaggerated in size, stands out as the central element of the scene. The child also includes himself in the drawing, depicted in a smaller size alongside his mother, both colored in black, occupying the central space of the image along the dirt track, representing the moment he was being carried. At the bottom of the drawing, the boy illustrates the route taken to receive medical care, using roads that connect cars to buildings, symbolizing the emergency care unit and the FMT-HVD. This spatial

separation between blocks of the image reinforces the transition from the familiar setting of the incident to the urban/hospital environment of care. The drawing features vibrant colors, simple lines and recognizable elements, showing that the child represented and organized his experience in a sequential and understandable way. The style of the drawing is classified as schematic, appropriate for his age, with recognizable shapes such as houses, birds in the sky, trees, vehicles and characters. The child is present in the drawing, placed centrally, as is the snake. Movement, the route to the hospital, the means of transportation used, and the presence of the mother as a companion are all represented. However, the drawing does not depict pain, the injury caused by the bite, clinical symptoms or the medical care received.

Children who did not have prior experience with snakebites had more difficulty identifying and understanding the snakebite. Some showed initial resistance to going to the healthcare unit, requiring their parents to insist on taking them. Perhaps because they had never experienced a snakebite, others initially denied the possibility of such an incident, telling their parents they had stepped on a nail or been bitten by an ant. However, as the pain progressed and symptoms worsened, the parents ended up seeking medical help.

*"She [my mom] said we should go to the hospital, and then I said I didn't want to go... because it wasn't doing anything anymore, right? But we had to go, right?" (P2, 11 years old, male)*

*"I was standing like this, helping my dad, and then I just felt a sting — I even thought it was an ant. But when I looked, I saw it was a snake that bit me. When it bit, I pulled away" (P8, 11 years old, male)*

*"I was playing... playing by the door... but later it felt like a nail [referring to the snakebite]" (P9, 5 years old, male)*

**4) Use of therapeutic practices during the child's healthcare itinerary**

The most common popular care practices were washing the bite site with water, applying topical products, using painkillers purchased from pharmacies, and performing tourniquets. The use of the folk subsystem, such as consultation with religious leaders, shamans or other healers, was not observed (S4 File).

*""I put alcohol on it because, when mommy puts it on my knee, in a second, it's already healed... So, I took some, put it on my hand and rubbed it on" (P1, 9 years old, female)*

*"Grandma didn't wash it. Mom, she didn't... she just tied it up" (P12, 5 years old, female)*

*""He just wiped it with a cloth [referring to his father], then tied it up. Later, I woke up and he untied it because the bleeding had already stopped." (P16, 10 years old, male)*

*"She applied a homemade medicine [referring to his mother], with those things she knows how to make… She washed it with warm water, and she gave me a metamizole pill to take and I think also some aspirin" (P19, 9 years old, female)*

The care provided by healthcare professionals was also reported by some children, which included the insertion of a venous access line, cleaning of the bite site with antiseptic, administration of intravenous analgesics, and the use of saline solution and antivenom serum. Only one child recognized and mentioned the effectiveness of the treatment and the improvement of symptoms following medical and hospital interventions.

*"They brought some cotton and medicine. Then, they applied it to my leg and later poured something over it to clean it" (P5, 12 years old, female)*

*"They gave me medicine for the pain through the vein here" (P7, 8 years old, male)*

*"My pee has gotten a lot better, glory to God. Now it's just a little yellow" (P19, 9 years old, female)*

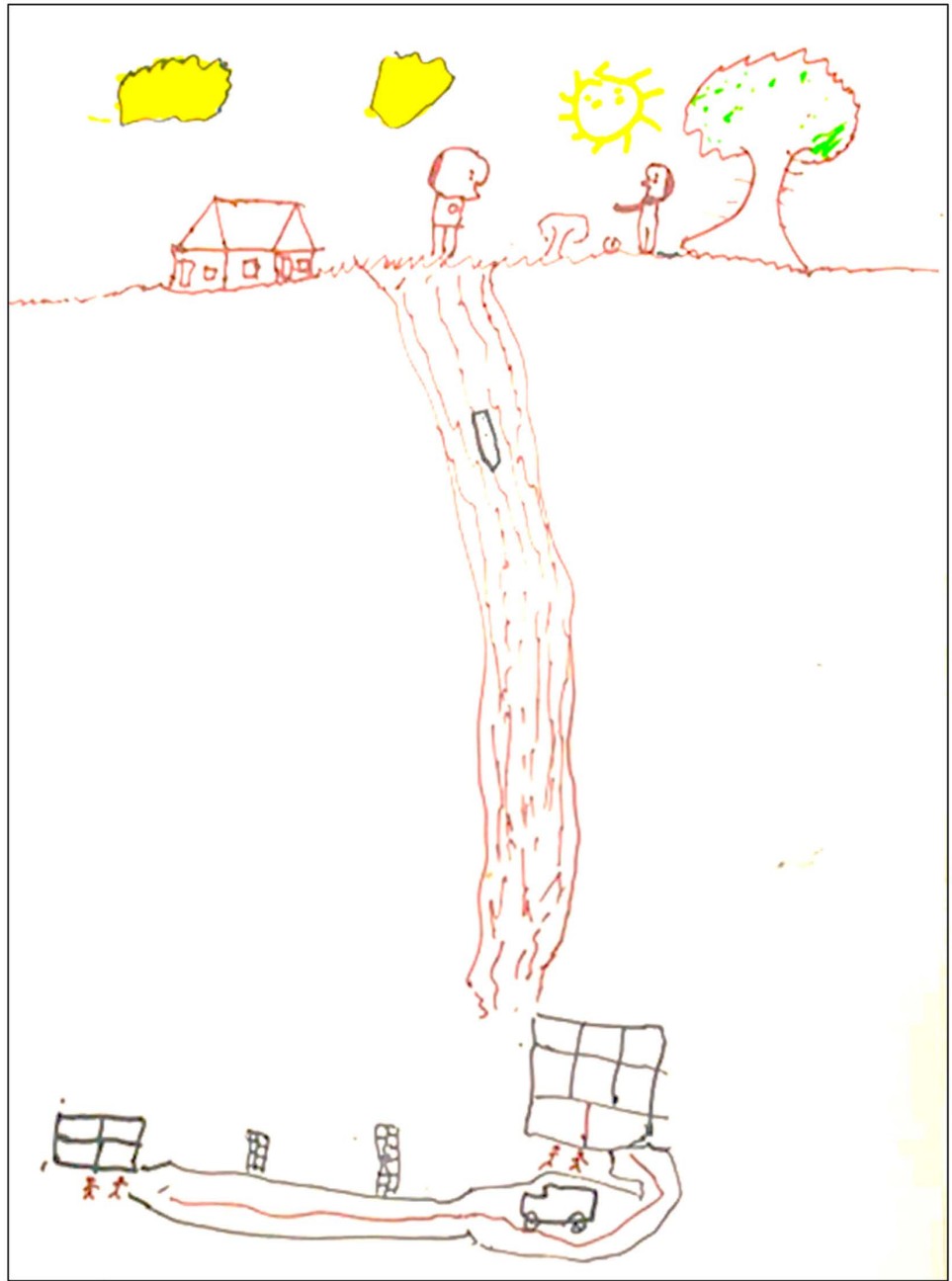

**Fig 6. Drawings made by participant 17 showing the journey undertaken.** The drawing portrays a snakebite experienced by an 11-year-old male child, a 6th-grade elementary school student. The incident occurred in the peridomestic area, in the backyard of his home, while the boy was harvesting olives. During the activity, he was bitten by a snake that was coiled near his feet, as depicted in the drawing. The child was accompanied by his father, who was close to him at the time. After the incident, the child was initially transported by motorboat to the port of the city of Rio Preto da Eva (Amazonas). From there, he traveled by private car to the municipal hospital. He was later transferred by ambulance to the Dr. Heitor Vieira Dourado Tropical Medicine Foundation (FMT-HVD) in Manaus. The drawing, created with realistic outlines and proportions appropriate for the child's age, presents a clear narrative structure, divided into three visual sections. In the upper part, the domestic environment is shown, with the house on the left, a tree on the right, and two human figures: the boy harvesting olives near the tree and the snake drawn coiled near his feet. The father is also depicted in the scene, accompanying the activity. The setting is characterized as daytime, with a visible sun and clouds in the sky. In the middle section of the drawing, a strip with a waterway connects the top and bottom parts, symbolizing the boat transport to the port. In the lower portion of the composition, the urban setting appears, with winding roads, a car, an ambulance, and buildings representing the municipal hospital and the FMT-HVD. There are also small human figures near the hospital buildings, representing the healthcare professionals who received and attended to the child. The drawing demonstrates movement

and a sequence of scenes, connecting the site of the incident to the path towards care and treatment. The style is classified as realistic, appropriate for the child's age of 11, showing proportional use of space, an attempt at perspective, realistic textures, colors and emotional expression, including the child depicted as screaming, representing the pain felt during the incident. The snake is drawn small and placed peripherally, as is the child, who is shown at a smaller but proportional scale. The route taken, the means of transport used, the presence of the father as a companion, the hospitals involved and the care provided are all clearly represented. Although the scene of pain is depicted, there is no direct illustration of the wound or systemic symptoms.

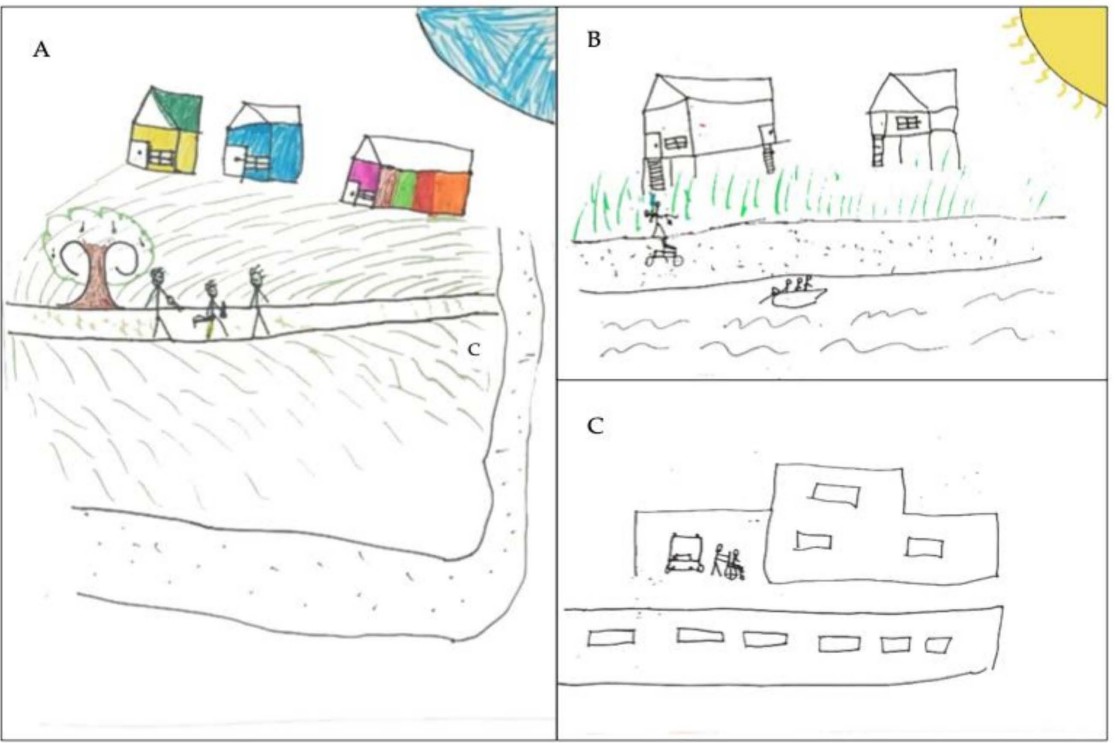

**Fig 7. Storyboard by participant 16 depicting his journey.** The drawings depict a snakebite experienced by a 10-year-old male child, a 4th-grade elementary school student. The incident occurred in the peridomestic area **(A)**, as the child was returning home at night, accompanied by his father and uncle. On the following day **(B)**, the child crossed the river in a small, motorized boat (riverboat) and was later transported by private car (C) to the municipal hospital in Iranduba (Amazonas). After evaluation, he was transferred by ambulance to the Dr. Heitor Vieira Dourado Tropical Medicine Foundation (FMT-HVD), where he received specialized care. The set of drawings produced by the child is organized in comic strip format, with three panels sequentially illustrating the most significant events of the episode—from the moment of the bite to the arrival at the hospital. In the first panel **(A)**, the scene of the incident is depicted, with a night sky and visible moon, indicating that the event took place at night. The child's house appears in the background, representing the destination he was heading to. In the second panel **(B)**, now in daylight—evidenced by the presence of the sun—the child is drawn being carried by his father and transported by motorcycle to a riverboat, demonstrating the variety of transportation methods used in the search for medical help. Finally, in the third panel **(C)**, now in the urban area, the boy is shown arriving at the hospital in his mother's car and being taken in a wheelchair into the healthcare facility. The drawing style is classified as realistic, appropriate for the child's age, showing attention to detail, an attempt at perspective, and the use of true-to-life colors. The participant is placed in a central position in the scene and is drawn at a proportional size. Although the snake is not depicted, the focus is on the sequence of events and the transportation methods used to reach medical care. Clear representations of movement, the vehicles used (motorcycle, riverboat, car and ambulance), the companions and the hospital are present, including the moment of arrival in a wheelchair, demonstrating the child's awareness of the care received. Despite the rich narrative and well-structured timeline, the drawing does not directly portray the injury, expressions of pain, the snake itself or physical symptoms.

## 5) Contingencies during the children's healthcare itinerary

The children's healthcare itinerary was marked by unexpected situations that further hindered their access to care. Geographic isolation was widely evident in the children's drawings and accounts, revealing a clear distance from medical assistance due to the long routes traveled (Fig 7). In addition, upon arriving at the health units near their homes or at hospitals in their municipalities in the interior of the state of Amazonas, they frequently encountered a lack of antivenom serum. Some children were left waiting at a single location for transportation to continue their journey. This waiting time ranged from a minimum of 30 minutes to a maximum of 72 hours, in other words, some cases were resolved quickly, but in others, the wait lasted days to secure transport for hospital transfer. Seven of the children, upon arriving at the nearest health unit and realizing, along with their families, that there were no available resources, and that the hospital transfer would take too long, sought their own means of transport until they finally reached a referral center with appropriate treatment.

Half of the interviewed children were alone at the time of the incident. Although there was an adult nearby, the children were unsupervised at that moment. Parents only went to the child after being approached for help or hearing their crying. In two cases, immediate assistance was provided by another child: in the first (P5), a child helped carry the victim to a safe place; in the second (P6), the victim's brother, also a child, removed them from the danger zone and went back to kill the snake.

*"My brother [another minor, 14 years old] killed the snake and called my mom [...] Then mom arrived, and she took me to the hospital, and that was it." (P6, 7 years old, male)*

*"I was alone crying... I didn't stay near it anymore [referring to the snake]" (P9, 5 years old, male)*

*"I went to change clothes [by myself], already at the end of the trail [...] when I was just coming out of the forest, something bit my leg, and then I just saw it bleeding." (P13, 12 years old, male)*

*"I was playing alone... I cried when it bit me [referring to the snakebite]." (P11, 6 years old, female)*

## Discussion

This study, through drawings and narratives of therapeutic itineraries, revealed unique insights into how SBEs are perceived and remembered by children. Most SBEs occurred in rural peridomestic settings, typically during routine daily activities. The health itineraries were characterized primarily by decisions made by caregivers and health professionals, involving the popular and professional subsystems. Only two children presented a mixed itinerary, actively participating in the beginning of their trajectory to care. The narratives and drawings highlighted intense pain, fear, and communication barriers with adults, highlighting children's vulnerabilities in the context of SBE emergency care.

Data triangulation, which combines different collection methods, has emerged as an innovative approach in research involving children, and provides a deeper and more reliable understanding of their experiences. The use of drawings allows overcoming cognitive and communicative limitations [39,40]. Interviews complement this approach by providing a direct account of their perceptions, while medical records add relevant clinical or historical context [41]. This methodology enhances the validity of the findings, as the convergence of multiple sources reduces bias and strengthens the credibility of the research [42]. Furthermore, by integrating these different approaches, researchers can capture the complexity of children's experiences in a more sensitive and ethical manner [43].

Drawing is an effective methodological tool for accessing children's perceptions and experiences in health contexts [44]. The use of drawing in this study helped to complement and reaffirm information about the children's health itinerary by allowing access to the child's emotional world in a non-verbal way [45]. Drawings served as a bridge between the objective account of health itineraries and the subjectivity of children's experiences, offering a deeper and more

empathetic understanding of their journeys [46]. Therefore, the incorporation of drawings into studies on health itineraries represent a promising methodological approach that values children's voices and perspectives.

Based on the anthropological framework of Tim Ingold, one may also adopt the notion of "itineration" to better understand the fluid, improvisational, and relational nature of therapeutic itineraries. This perspective highlights that therapeutic care in health does not occur in a linear or rigidly structured manner but rather emerges from multiple, overlapping, and dynamic movements shaped by emotional, sociocultural, and contextual factors [47]. The experience of the journey to seek care after a snakebite can also be understood considering the concept of 'non-places', as proposed by Marc Augé [48]. Environments such as health centers, impersonal waiting rooms or long journeys constitute transitional spaces that, do not favor identity bonds, acceptance or the construction of 'narratives of belonging' [48]. These non-places highlight the emotionally destructive dimension of the therapeutic itinerary, especially in moments of vulnerability.

Moreover, children were often without direct adult supervision or accompanied only by other minors at the time of the bite, which may have been a contributing factor in delaying immediate mobilization and aid. Another important factor was that, although parental care remained predominant in decision-making throughout the children's therapeutic itinerary, there were episodes of spontaneous self-care, with the use of homemade remedies learned through observation of adult behavior. Therefore, while the adult therapeutic itinerary is heavily impacted by structural and geographical limitations [14], the children's itinerary is additionally influenced by relational, emotional and socio-cognitive factors, which hinder the early recognition of potential severity by caregivers [8].

## Environmental and sociocultural factors in children's healthcare itinerary

This study showed that most SBEs occurred near the children's homes, often while they were engaged in daily activities, playing outdoors, or helping their parents in labor activities. Snakebites are intrinsically linked to the everyday activities of rural populations, especially when associated with agricultural work and the exploitation of natural resources [49]. Children living in rural areas or assisting with agricultural work are among those most affected by SBEs and, in many cases, may face permanent physical disabilities, compromising their quality of life [50].

A sense of trust and safety regarding the area around their homes was noted among the children, which can lead to underestimation of the risk and a lack of protective measures. In the study by DooKeeram et al. [49], the lack of use of protective measures was also attributed to low-risk perception. When identifying this low perception of risk, health education stands out as an essential strategy for preventing accidents in childhood, especially in contexts of high social vulnerability. According to Laguna et al. [51], educational interventions aimed at children should be implemented in conjunction with strategies aimed at guardians or caregivers. In Sri Lanka, persistence of harmful practices among caregivers revealed an urgent demand for culturally adapted programs to improve access to treatment for SBEs in children [52]. Caregivers with prior access to information or first-aid training engaged in significantly fewer harmful practices [53]. Community education programs developed in the Brazilian Amazonia increased the population's knowledge about SBE prevention and encouraged early medical care [54]. In the same direction, the WHO report recommends the inclusion of prevention content in school curricula and the involvement of teachers and community leaders as fundamental strategies to reduce SBE-associated morbidity and mortality [2]. Preventive interventions targeting SBEs among children in the Brazilian Amazon need to be implemented.

In this study, a sociocultural pattern observed was an adults' initial disbelief in the children's account of an SBE. Grills and Ollendick [55] postulated that there is a tendency to place greater value on adult reports, and to consider that children are unable to coherently or accurately describe traumatic events. This results in poor agreement between the child's and the caregiver's quotes, with a tendency for adults to seek alternative explanations before accepting the child's report as truthful [56].

Another practice observed in this study was self-care by the children and the phenomenon of children caring for other children. This has been discussed in the literature as a complex expression of adaptation in the face of social and

economic vulnerability. The study by Poletto et al. [57] highlights that girls between aged eight and twelve, living in impoverished communities, often assume significant responsibilities in caring for younger siblings and managing household tasks. Cunha [58] warns that child-to-child caregiving within the domestic sphere can constitute a form of child labor, particularly when it involves overload and lack of adult support. A critical reflection on these practices requires consideration of the historical, social, cultural, and financial determinants that shape family dynamics in different communities. Therefore, while caregiving among children may, in certain contexts, promote emotional development and resilience, careful attention must be paid to the conditions under which these practices occur, so they do not become tools of exploitation or means of rendering childhood invisible [59].

In addition, it was observed that the behavior of the child and their caregiver in seeking help in an intuitive and disorganized manner, without clear guidance on the most appropriate destination for receiving medical care. Santos et al. [60] highlighted a sense of invisibility from populations living in remote areas of the Brazilian Amazon, with an absence of clear strategies to address emergencies such as SBEs.

The limitations of this study include the subjectivity of interviews, in which responses may be influenced by personal interests and hesitations, potentially affecting the analysis. The results reflect the specific reality of the individuals studied and are not intended for generalization. In addition, cultural and language barriers, especially among indigenous children, may have made it more difficult for them to express their experiences, potentially affecting the depth and clarity of the responses. Despite these limitations, the study provides valuable insights into the challenges faced by children who are victims of snakebites in the Amazon, highlighting the vulnerability of these populations. Future research should consider the use of the playful drawing method to explore the psychological and emotional impact of these experiences.

## Conclusion

This study revealed that children's therapeutic itineraries following SBEs in the Brazilian Amazon are shaped by a complex intersection of structural, sociocultural and emotional dimensions. The innovative use of thematic drawings and storytelling provided a unique window into children's lived experiences, enabling a more nuanced understanding of how they interpret and react to traumatic events. While geographic barriers and lack of antivenom in community health units remain critical bottlenecks to SBE care, the believe that children are unable to coherently describe their health status and limited adult supervision are determinants of SBE itineraries in our study setting. These dynamics underscore the urgent need to reframe healthcare strategies from a child-centered perspective, recognizing children as active agents whose experiences must inform health policies.

Improving the care of SBEs in children in the Amazon demands culturally sensitive, developmentally appropriate and participatory strategies to engage families, communities and healthcare providers in recognizing early symptoms, validating children's accounts and reducing delays in accessing professional care. By doing so, it is possible to transform fragmented and improvised therapeutic itineraries into structured and responsive care pathways to proper and timely treat SBEs in children in the rural Amazon.

## Supporting information

**S1 File. Consolidated criteria for reporting qualitative research.**
(DOCX)

**S2 File. Characterization of the children who were victims of snakebite accidents and participated in the study.**
(DOCX)

**S3 File. The environment and participant activities at the time of the snakebite.**
(DOCX)

**S4 File. Use of the popular healthcare sector in the therapeutic itineraries of the study participants.**
(DOCX)

**S5 File. Children's drawings depicting snakebite envenomation: a descriptive analysis.**
(DOCX)

## Author contributions

**Conceptualization:** Joseir Saturnino Cristino, Altair Seabra Farias, Erica Silva Carvalho, Jacqueline Sachett, Fan Hui Wen, Felipe Murta, Vinícius Azevedo Machado, Wuelton Monteiro.

**Funding acquisition:** Wuelton Monteiro.

**Investigation:** Joseir Saturnino Cristino, Vinícius Azevedo Machado.

**Methodology:** Jacqueline Sachett, Vinícius Azevedo Machado, Wuelton Monteiro.

**Project administration:** Jacqueline Sachett, Vinícius Azevedo Machado, Wuelton Monteiro.

**Supervision:** Vinícius Azevedo Machado, Wuelton Monteiro.

**Writing – original draft:** Joseir Saturnino Cristino, Fan Hui Wen, Felipe Murta, Vinícius Azevedo Machado, Wuelton Monteiro.

**Writing – review & editing:** Joseir Saturnino Cristino, Altair Seabra Farias, Erica Silva Carvalho, Jacqueline Sachett, Fan Hui Wen, Felipe Murta, Vinícius Azevedo Machado, Wuelton Monteiro.

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
