## [Decision Letter · Decision Letter 0]

2 Sep 2025

Therapeutic Itineraries of Children after Snakebites in the Brazilian Amazon: a Thematic Drawing-and-Story Study

Dear Dr. Wuelton Monteiro

Thank you for submitting your manuscript to PLOS Neglected Tropical Diseases. After careful consideration, we feel that it has merit but does not fully meet PLOS Neglected Tropical Diseases's publication criteria as it currently stands. Therefore, we invite you to submit a revised version of the manuscript that addresses the points raised during the review process.

Please submit your revised manuscript within 60 days, with the REVISION DUE on 30th October 2025. If you will need more time than this to complete your revisions, please reply to this message or contact the journal office at plosntds@plos.org. Please include the following items when submitting your revised manuscript:

We look forward to receiving your revised manuscript.

Kind regards,

SELORME ADUKPO

Guest Editor

José María Gutiérrez

Section Editor

Shaden Kamhawi

co-Editor-in-Chief

Paul Brindley

co-Editor-in-Chief

**Additional Editor Comments:**

Reviewer #1:

Reviewer #2:

Reviewer #3:

**Journal Requirements:**

1) We do not publish any copyright or trademark symbols that usually accompany proprietary names, eg ©,  ®, or TM  (e.g. next to drug or reagent names). Therefore please remove all instances of trademark/copyright symbols throughout the text, including:

- ® on page: 8.

2) Some material included in your submission may be copyrighted. According to PLOSu2019s copyright policy, authors who use figures or other material (e.g., graphics, clipart, maps) from another author or copyright holder must demonstrate or obtain permission to publish this material under the Creative Commons Attribution 4.0 International (CC BY 4.0) License used by PLOS journals. Please closely review the details of PLOSu2019s copyright requirements here: PLOS Licenses and Copyright. If you need to request permissions from a copyright holder, you may use PLOS's Copyright Content Permission form.

Potential Copyright Issues:

- Please confirm (a) that you are the photographer of Figures 3, 4, 5, 6, 7, and and S2., or (b) provide written permission from the photographer to publish the photo(s) under our CC BY 4.0 license.

**Reviewers' Comments:**

Reviewer's Responses to Questions

**Key Review Criteria Required for Acceptance?**

**Methods:**

-Are the objectives of the study clearly articulated with a clear testable hypothesis stated?

-Is the study design appropriate to address the stated objectives?

-Is the population clearly described and appropriate for the hypothesis being tested?

-Is the sample size sufficient to ensure adequate power to address the hypothesis being tested?

-Were correct statistical analysis used to support conclusions?

-Are there concerns about ethical or regulatory requirements being met?

Reviewer #1: This is a qualitative research study that includes quantitative clinical data and qualitative data from interviews that included drawing and narratives. The data collection tools (epidemiologic and TD-SP) are clearly described as are the methods used for managing and analyzing the data (transcription, descriptive statistics, coding, and content and thematic analysis). The study was overseen by an IRB committee and both parental and child participant acceptance was obtained.

The following issues also need to be addressed:

- Please provide more detail on how participants were selected and recruited (e.g., was it every child presenting with snakebite during the study period?) and 2. how they were recruited for the study (e.g. a point person in the hospital alerted the study team who then approached the parent first and child second to explain the study and ask if they were willing to participate)? Clarify also whether all participants approached agreed to participate.

- How was the sample size determined? If saturation was used how was this determined?

- Re the first paragraph under ‘Data analysis’ in the methods section - It is not clear what this is about. How does "classification" relate to coding and theme identification described below?

Consider clarifying this section, and if it does not add substantially to the analysis, it may be best removed.

- Clarify in the last paragraph of the methods section if ‘categories’ are themes.

- The methods section states that participants were told their drawings would not be shared yet these are included in the paper.

Reviewer #2: This is a unique study and it is therefore not easy to simply answer some of these questions. This is a mixed-methods analysis of childrens' experiences with snakebite and seeking care for snakebite.

The study population is well-described. Not all components of the analysis are hypothesis driven, but I believe that is appropriate, as it is partially an exploratory and descriptive analysis.

Reviewer #3: The objective of the study was to explore the therapeutic itineraries of 20 children following snakebites in the Brazilian Amazon, using a participatory approach based on their drawings and narratives. The study design is clearly stated; the population and data collection are described in detail. Methodological triangulation was used for robustness. Ethical approval has been obtained.

- “No interviewer-related biases were identified”– how did you minimize/ check for interviewer bias?

- Was member-checking performed?

- Potential power dynamics and cultural considerations should be addressed in the researcher’s reflexivity

- Please, add information on sample size, e.g. how was data saturation defined?

- Please clarify how data was analyzed. In the abstract it is stated “Data was analyzed by deductive content analysis”. In the method section it’s stated that “A coding system was adapted from Goldner et al.“- applying deductive coding, and inductive coding was applied for narratives. Was a hybrid/ mixed approach used?

- “The results and methods report follow the Consolidated Criteria for Reporting Qualitative Research“ (S1 file). – The uploaded S1 file does not show COREQ guidelines, it shows Table 1. Characterization of the children who were victims of snakebites and participated in the study.

**Results:**

-Does the analysis presented match the analysis plan?

-Are the results clearly and completely presented?

-Are the figures (Tables, Images) of sufficient quality for clarity?

Reviewer #1: Overall the results are presented in detail. The themes are especially well developed pulling from multiple sources and illustrated with quotes. In the first section (before themes) there is some repetition and I think some of the detail can be reduced - more detail is included in my comments below.

- Details presented in Tables 2 and 5 can be moved to a supplement file with key examples extracted to illustrate main findings.

- Table 2: P2 is “foot” an accurate translation? Based on context “boot” seems more likely. And picking fruit in the tree is classified as Workplace for P2 and peridomestic for P4

- What is meant by active vs passive itineraries?

- Para starting “Table 3 presents the classification..” ends with this sentence “The following decisions were made by their companions.” However this is not followed by any more information on the decisions made.

- Para starting “Seventeen children…” states one child had mixed itinerary - and then goes on to give examples of 3 participants (P 5, 6 and 19) who seem to have mixed itineraries

- Para starting "Eighteen (90%) participants…” - what is the difference between ‘multidimensional’ and ‘fragmented’?

- Para starting "Eighteen (90%) participants…”- There appears to be some repetition in this section. Consider streamlining by only briefly referring to Table 3 here and addressing the details more fully under themes

- Table 3 would benefit from an additional column that explains the categories in more detail e.g. what is meant by ‘decision maker’? (decision on what?); what is fragmented vs continuous care?; What are other classification of severity?; if only 20% are public providers, what are the others?; what is meant by itinerary reporter?, etc

- Is there any information that can be shared on where the 4 participants who receive antivenom before being admitted to the hospital got it from?

- ‘Drawing’ analysis section - Both the descriptive paragraph and the table communicate similar findings—consider retaining one and streamlining the other.

Reviewer #2: The results do match the analysis plan.

The results are clear, but there is too much text and description. The results need to be better condensed and need to be presented in a much more concise manner. Things that the authors want to present that don't make it into the main paper can be presented in the supplemental files. As is, the paper reads more like a dissertation than a research paper.

Figures are of sufficient quality and clarity.

Reviewer #3: Overall, the presentation of the results could be improved. Reduce redundant information in both text and tables to improve clarity. The manuscript would benefit from including not only descriptive analysis of the data but also a critical interpretation of the drawings and narratives to provide deeper insight into the participants’ experiences.

The topics of challenges in understanding the child’s narrative, perceptions of severity (e.g. hematuria), and health literacy, as reasons for delayed care-seeking are highly interesting.

Tables & Figures:

Include only necessary tables and figures in the main text; additional materials should be provided in the supplementary files.

- e.g. Figure 3-7: are duplicates and are already included in the supplementary materials S2

- Figure 2: Consider revising the data presentation format to enhance clarity and readability

- Table 3 and 5: reporting only descriptive data by individual participants may reduce readability and analytical value. Consider including critical analysis to highlight boarder patterns and insights.

**Conclusions:**

-Are the conclusions supported by the data presented?

-Are the limitations of analysis clearly described?

-Do the authors discuss how these data can be helpful to advance our understanding of the topic under study?

-Is public health relevance addressed?

Reviewer #1: The conclusions are clear and highlight the study's main findings. They are mostly well grounded in study findings. The sentences towards the end on future public health interventions however need to be supported by more references to evidence in existing literature (included in the main body of the discussion) to complement current study findings.

Reviewer #2: The conclusions are supported by the data and also by other literature.

This is a small sample size, so it would be good to offer even more caution with regard to extrapolating to other similar individuals, let alone populations.

There is some public health relevance here, and it is an important topic. I do believe a stronger public health argument could be made. There is clear public health relevance, but some of the text seems to get lost in social science theory rather than in practical, applied importance of the findings and what they might mean for public health etc.

Do keep in mind that some aspects of what are being described are culture-specific and may also be symptoms of poverty. The authors note that such things must be taken into account by public health authorities. Children not being supervised, for example, might be quite relative across cultures. The paper almost tends toward casting blame for this when it could be culturally construed and/or a symptom of poverty.

Reviewer #3: Discussion

Consider reporting the key findings in the first paragraph of the discussion to provide context for the reader before interpreting them.

“In the case of the children in this study, initial delays were often related to adults’ disbelief in the child’s account of the incident, difficulties in understanding the child’s narrative, and the consequent postponement of seeking professional care.”

- This is a very important finding, how could this be improved? Are there any strategies or educational interventions in place for children in rural communities, who are at risk of snakebites?

- Does Brazil have established guidelines for first aid and healthcare-seeking pathways following snakebite envenoming, and how do communities at risk obtain this information?

The limitations are stated.

Conclusion

In the abstract, the conclusion could be made more specific by highlighting how the findings might inform or enhance educational interventions for children and caregivers. Refining the abstract conclusion would also highlight the public health relevance of the study.

**Editorial and Data Presentation Modifications?**

Reviewer #1: Introduction:

The introduction starts very clearly providing context on snakebites globally and in Brazil.

The concept of “healthcare itinerary" is critical to this study and required a clearer introduction and definition, with reference to literature on this approach. This could largely be achieved by revising the third paragraph.

While interesting, too much historical information is provided on the history of the study of children’s perspectives - this would be better summarized in one paragraph that justifies why studying the child’s unique perspective is important..

Methods:

- I find the paragraph that starts “In this study, a healthcare itinerary…” hard to follow and I am not sure what the purpose of having it in the methods section is.

- Consolidated Criteria for Reporting Qualitative Research should be cited- not attached as a supplement.

Discussion:

The discussion is currently the weakest element. It is overly long in parts, and could be sharpened by focusing more directly on the key findings and situating them in the wider literature.

- The flow would be improved by starting with the key findings and setting it in context of the literature - and only then including one paragraph (not 5 or 6) on why this method is important - it will be evident at that point the added value that this approach has provided.

- Each of the findings on the unique child child perspective identified in this study should be presented with a one sentence summary of the result and then setting it in context of the wider public health literature (beyond snakebites). E.g "child disbelief in adult account’ - how does this show up in studies on other public health issues? For the finding of a sense of trust around the house/ work you cite DooKeerem - has this phenomenon also been reported in other public health studies?; etc. This was done well for the finding ‘understanding child’s initial account’ where findings from Grills and Olendick were referenced.

- How data showed up in the drawing is another important section of this discussion and substantial reference is appropriately made to the literature on this. That paragraph needs to start with one sentence summary of the relevant findings from this study - before unpacking and interpreting.

- The in depth discussion on child labor while interesting is less relevant to this paper.

Reviewer #2: There are some points that could be made more clear in the writing, but I believe the entire manuscript needs to be made vastly more concise before it is worth pointing out smaller grammatical edits, etc.

Reviewer #3: (No Response)

**Summary and General Comments**

Reviewer #1: This is an important and novel study that uses detailed methods to illuminate children’s perspectives on snakebite care, with valuable implications for the design of public health interventions. At the same time, the manuscript is overly long in places, lacks focus in the discussion, and omits some key methodological details. Addressing these issues will substantially strengthen the clarity and impact of the paper.

Overall the writing is good but there are some sentences and paragraphs where the meaning is not clear to me. The paper would benefit from being reviewed by an editor.

Reviewer #2: I strongly suggest making the entire manuscript much more concise. There is background materials touching on the history of different social science approaches, etc. and much of this is both not sufficiently detailed to pass social science writing AND not really necessary for this public health journal. The writing also seems to have difficulty in its direction. Is this a methods piece, meant to showcase this style of triangulation? Or is this a study about snakebite and snakebite-related care among children? I don't believe it should be both, even though it could be a paper about snakebite and snakebite-related care, which showcases a novel approach to data collection and analysis (this is how I would frame this paper).

The results also need to be condensed. Try to make one point = one paragraph at most, and then with only 1-2 quotes per point.

Finally, in the discussion - keep this to being a paper about snakebite and snakebite related healthcare as a problem that needs to be addressed. What specifically did we learn? Is it similar or different from similar or very different settings? What are concrete public health things that we take from this. Try to be as concise as possible - this is different from normal social science writing. Most readers will not make it through something so long, and with as much materials about social science history and theory. The goal should be to make this so that it is broadly accessible (including easy to read and get important points from it) - and perhaps especially for folks who might be in a position to deal with this problem.

Reviewer #3: The manuscript contributes valuable insight into a neglected area of research. This study explores the therapeutic itineraries of 20 children following snakebites in the Brazilian Amazon, using a participatory approach based on their drawings and narratives combined with clinical and epidemiological data. The findings explore the children’s experience but also show how caregivers’ disbelief in the child’s account of the incident, perception of severity, and health literacy impact the delay in seeking professional care.

This study highlights the need to enhance health literacy in at-risk communities. This may influence timely care-seeking, adherence to first aid guidelines, and interpretation of children’s experiences following snakebite envenoming.

The presentation of the results could be improved. Reduce redundant information in both text and tables to improve clarity. The manuscript would benefit from including not only descriptive analysis of the data but also a critical interpretation of the drawings and narratives to provide deeper insight into the participants’ experiences.

Cave: using participants numbers in citations and text may breach confidentiality, particularly in this small sample size and context-specific study. In addition, assigning participant numbers to detailed clinical data and drawings could compromise confidentiality (Supplementary Information 1. Table 1. Characterization of the children who were victims of snakebites and participated in the study; and Supplementary Information 2. Drawings of participants). The authors should clarify how data were anonymized or consider aggregating findings to prevent identification.

Specific comments:

- Instead of the term “playful method”, the terms “child-friendly or age-appropriate method” could be used, this might be a more suitable term.

- The term “fragmented transport” should be clarified.

- “Through content analysis of the interviews with and drawings by participants,

five themes emerged“ – please, improve sentence structure.

PLOS authors have the option to publish the peer review history of their article (what does this mean? ). If published, this will include your full peer review and any attached files.

**Do you want your identity to be public for this peer review?** For information about this choice, including consent withdrawal, please see our Privacy Policy .

Reviewer #1: **Yes: ** Margaret C Baker

Reviewer #2: No

Reviewer #3: No

**Figure resubmission:**
---

## [Decision Letter · Decision Letter 1]

19 Nov 2025

Dear Dr Monteiro,

We are pleased to inform you that your manuscript 'Therapeutic Itineraries of Children after Snakebites in the Brazilian Amazon: a Thematic Drawing-and-Story Study' has been provisionally accepted for publication in PLOS Neglected Tropical Diseases.

Best regards,

José María Gutiérrez

Section Editor

Shaden Kamhawi

co-Editor-in-Chief

Paul Brindley

co-Editor-in-Chief

Reviewer's Responses to Questions

**Key Review Criteria Required for Acceptance?**

**Methods**

-Are the objectives of the study clearly articulated with a clear testable hypothesis stated?

-Is the study design appropriate to address the stated objectives?

-Is the population clearly described and appropriate for the hypothesis being tested?

-Is the sample size sufficient to ensure adequate power to address the hypothesis being tested?

-Were correct statistical analysis used to support conclusions?

-Are there concerns about ethical or regulatory requirements being met?

Reviewer #1: Just 2 outstanding issues remain:

1. The sentence “All the children included in the study were admitted at FMT-HVD for SBE treatment during the study period.” still does not make it clear to me what the selection strategy is. Maybe the authors meant “All the children admitted at FMT-HVD for SBE for treatment during the study period were included”? If not, how were they selected?

2. While the description on analysis has been simplified I am still unclear on the relationship between ‘categories’ and ‘themes’ used at the end of this section and how they were applied. This is how I am interpreting it for myself - but I am not sure if this is correct: Themes were identified through inductive content analysis and discussion among the researchers. Drawings and transcripts were then coded with these themes (manually or using software?).

Reviewer #2: (No Response)

Reviewer #3: All suggested changes have been discussed or implemented.

**Results**

-Does the analysis presented match the analysis plan?

-Are the results clearly and completely presented?

-Are the figures (Tables, Images) of sufficient quality for clarity?

Reviewer #1: (No Response)

Reviewer #2: (No Response)

Reviewer #3: No issues were identified; all suggested changes have been discussed or implemented.

**Conclusions**

-Are the conclusions supported by the data presented?

-Are the limitations of analysis clearly described?

-Do the authors discuss how these data can be helpful to advance our understanding of the topic under study?

-Is public health relevance addressed?

Reviewer #1: (No Response)

Reviewer #2: (No Response)

Reviewer #3: All suggested changes have been discussed or implemented.

**Editorial and Data Presentation Modifications?**

Reviewer #1: (No Response)

Reviewer #2: (No Response)

Reviewer #3: (No Response)

**Summary and General Comments**

Reviewer #1: I appreciate the major efforts made by the authors to revise this text, and their patience in aligning their writing style with the global public health discipline used by PLoS NTDs. It is very interesting to have a sociological and educational perspective presented on the control of this NTD.

Reviewer #2: The authors have sufficiently addressed my critiques. I think this is an interesting study and analysis and that it is worth publishing. It might be worthwhile to have a light editorial review of the language and the flow of the prose. Congrats to the authors.

Reviewer #3: All suggested changes have been discussed or implemented.

PLOS authors have the option to publish the peer review history of their article (what does this mean? ). If published, this will include your full peer review and any attached files.

**Do you want your identity to be public for this peer review?** For information about this choice, including consent withdrawal, please see our Privacy Policy .

Reviewer #1: **Yes: ** Margaret C Baker

Reviewer #2: No

Reviewer #3: No

---

## [Editor Report · Acceptance letter]

Dear Dr. Monteiro,

We are delighted to inform you that your manuscript, "Therapeutic Itineraries of Children after Snakebites in the Brazilian Amazon: a Thematic Drawing-and-Story Study," has been formally accepted for publication in PLOS Neglected Tropical Diseases.

Best regards,

Shaden Kamhawi

co-Editor-in-Chief

Paul Brindley

co-Editor-in-Chief
